# Tocotrienol-Rich Fractions Offer Potential to Suppress Pulmonary Fibrosis Progression

**DOI:** 10.3390/ijms232214331

**Published:** 2022-11-18

**Authors:** Yifei Lu, Yihan Zhang, Dengfeng Xu, Yuanyuan Wang, Da Pan, Pei Wang, Jiayue Xia, Shiyu Yin, Wang Liao, Shaokang Wang, Guiju Sun

**Affiliations:** Key Laboratory of Environmental Medicine and Engineering of Ministry of Education, Department of Nutrition and Food Hygiene, School of Public Health, Southeast University, Nanjing 210009, China

**Keywords:** pulmonary fibrosis, TRF, carotene, combined therapy

## Abstract

Although pulmonary fibrosis (PF) is considered a rare disease, the incidence thereof has increased steadily in recent years, while a safe and effective cure remains beyond reach. In this study, the potential of tocotrienol-rich fractions (TRF) and carotene to alleviate PF was explored. PF was induced in Sprague-Dawley rats via a single intratracheal bleomycin (BLM) (5 mg/kg) instillation. These rats were subsequently treated with TRF, carotene, pirfenidone (Pir) and nintedanib (Nin) for 28 days via gavage administration, whereafter histopathological performance, biochemical functions and molecular alterations were studied in the lung tissues. Our results showed that TRF, carotene, Nin and Pir all ameliorated PF by reducing inflammation and resisting oxidative stress to varying degrees. The related mechanisms involved the TGF-β1/Smad, PI3K/Akt and NF-κB signaling pathways. Ultimately, our findings revealed that, when combined with TRF, the therapeutic effects of Nin and Pir on PF were enhanced, indicating that TRF may, indeed, provide promising potential for use in combination therapy in the treatment of PF.

## 1. Introduction

The 2019 outbreak of the coronavirus disease (COVID-19), a communicable respiratory disease caused by the SARS-CoV-2 virus, developed rapidly into a severe and currently ongoing global health challenge [1]. According to the World Health Organization (WHO), as of June 2022 more than 530 million people worldwide had reportedly been infected with the original SARS-CoV-2 or its variants of which over 6.3 million had died. Many of those who recovered from COVID-19, particularly those who suffered severe infection, are reported to be at risk of long-term pulmonary complications [2]. Interstitial pneumonia is a common feature of COVID-19 and can be complicated by acute respiratory distress syndrome (ARDS) of which pulmonary fibrosis (PF) is the recognized sequelae [3]. Autopsy results of COVID-19 patients have shown severe fibrosis in the lungs after an extended course of the disease [4]. Consequently, due to the severity and continuity of the epidemic and the burden of PF after SARS-CoV-2 infection, the management and amelioration of the disease is an increasing topic of medical and scientific discussion.

PF, indicated by the scarring of the lungs, is the ultimate and most devastating consequence of various inflammatory lung diseases [5]. While generally a rare disease, in recent years, numerous global studies have reported an alarming increasing trend in the incidence and prevalence of PF [6]. Unfortunately, medical theories and research results have not yet clarified the cause and mechanism of its pathogenesis. Glucocorticoids and immunosuppressive drugs are currently the main response to the disease in China. However, long-term use induces side effects, such as infection and bone necrosis, and large-scale use risks the development of liver necrosis. Moreover, these drugs are only effective in some patients with PF [7]. In October 2014, the U.S. Food and Drug Administration (FDA) approved pirfenidone (Pir) and nintedanib (Nin) specifically for the treatment of PF. However, these two drugs can only slow disease progression and pulmonary function damage in mild and moderate patients. They cannot completely prevent or reverse PF, and their efficacy and safety in patients still require in-depth exploration [8,9]. At present, the most effective treatment for PF is lung transplantation [10]. However, this approach is challenged by the limited supply of organ resources and donor institutions, the subsequent risks of infection, rejection and complications, with median post-transplantation survival only 4.5 years, the high cost of the procedure and the lifelong need for immunosuppressive therapy, all of which have limited its clinical promotion and application [11]. Therefore, the exploration of the pathogenesis of PF, determination of effective therapeutic targets and development of therapeutic drugs with stable curative effects and minimal side effects are far-reaching issues requiring urgent attention in the clinical and scientific research work of the medical community to reduce mortality and improve the prognosis of patients with PF.

Vitamin E, a potent antioxidant, is generally divided into two major categories, namely tocopherols and tocotrienols, of which tocotrienols have been shown to have more potent anti-inflammatory and antioxidant properties [12,13]. In terms of anti-inflammatory activity, tocotrienols inhibit the expressions of TNF-α, IL-1, IL-6, IL-8, inducible nitric oxide synthase and cyclo-oxygenase [14]. Tocotrienols have also been shown to inhibit the inflammatory response by inhibiting the STAT3 cell signaling pathway [15], and to prevent diabetes-related behavioral, biochemical and molecular changes by inhibiting the activation of the NF-κB signaling pathway [16]. In terms of antioxidant activity, tocotrienols have been shown to maintain normal physiological processes by regulating and scavenging the balance of reactive oxygen species (ROS) in the system [17]. Tocotrienol-rich fractions (TRF) extracted from palm oil are frequently used in tocotrienol research [18,19] and, hence, are included among the samples in this current work. Furthermore, vitamin A (retinol) is an important nutrient for lung, heart and liver tissues [20] of which the lung is the most sensitive to its effects [21]. The phytonutrients α- and β-carotene are commonly used as precursors for vitamin A synthesis. Carotene also plays a role in the management of oxidative stress, which stems from infection and inflammatory processes [22] and may contribute to diffuse lung disease (DLD) [23]. Epidemiological studies have shown that the incidence of various malignant tumors is lower in populations whose diets are rich in β-carotene or carotenoid-rich fruits and vegetables. Moreover, multiple cell and animal experiments have found that β-carotene has a significant anti-cancer effect, which is closely related to the antioxidant and immune-enhancing functions of β-carotene. Other studies have shown that β-carotene may play an important role in the release of lipids, such as prostaglandins, with immunomodulatory effects, thereby regulating the body’s immune function [21,24]. These beneficial functions of tocotrienols and carotene could significantly impact the occurrence and development of PF. Therefore, this study investigated the relationship between PF and the interventions of TRF and carotene and explored the potential of their combined therapeutic effects on the disease.

## 2. Results and Discussion

### 2.1. Effects on Weight (g), Feed Intake (g) and Lung Index of TRF

As can be seen from Figure 1, there were two significant fluctuations in the body weight of the rats during the 56-day experimental period. The first fluctuation occurred when the model was established on the first day, when the body weight of the rats in each group decreased significantly, thereafter recovering after 7–8 days (Figure 1a). The second fluctuation was observed when the gavaging began on the 28th day, after which the weight stabilized for 2–3 days and then recovered again (Figure 1a). Correspondingly, during these periods of weight loss, feed intake reduced accordingly (Figure 1b). The Nin group had the lowest body weight, corresponding to the lowest feed intake. The lung/body weight ratio (lung index) is an acknowledged index of PF; however, in this experiment, the lung indexes of each group were not statistically different (Figure 1c).

### 2.2. Effects on HYP and MMP-7 Level of TRF

As shown in Figure 2, high levels of HYP and MMP-7 contents indicated successful modeling of PF in the model group. The contents of HYP and MMP-7 in the model (BLM) group were significantly higher than those in the control group, rising from 0.11 ± 0.02 and 0.61 ± 0.02 to 6.55 ± 0.28 and 23.40 ± 1.75 mg/g, respectively, in the wet tissue, reflecting significant increases of 455% and 257%, respectively. After treatment with TRF and carotene groups and their combined group, the values of HYP and MMP-7 were reduced, but the therapeutic effect was significantly lower than that of the other drug groups (Nin, Nin + TRF, Pir and the Pir + TRF groups). Among the groups, only the TRF group was statistically different to the corresponding model groups in terms of both HYP and MMP-7 values (*p* < 0.05). The best therapeutic effects were seen in the Nin + TRF group and PIR group, which decreased by 110% (from 0.61 ± 0.02 to 0.29 ± 0.03 mg/g wet tissue, *p* < 0.05) and 82% (from 23.40 ± 1.75 to 12.85 ± 1.12 mg/g wet tissue, *p* < 0.05), respectively. While combined with TRF, the value of HYP and MMP-7 in Nin group and Pir group decreased except the Pir group in MMP-7, which showed better antifibrotic characteristics after added TRF. 

### 2.3. Effects on Inflammatory Markers of TRF

As shown in Figure 3, the values of IL-1β, IL-6, MPO, TGF-β1 and TNF-α rose from 169.00 ± 8.12, 950.60 ± 159.40, 46.89 ± 2.21, 772.20 ± 69.27 and 68.12 ± 3.40 mg/g protein in the control group, respectively, and to 394.50 ± 20.69, 3353.00 ± 586.90, 150.90 ± 9.62, 1744.00 ± 132.70 and 213.20 ± 36.05 mg/g protein (*p* < 0.05) in the model group, respectively. The TRF, carotene and drug groups all showed excellent anti-inflammatory abilities. The greater effects, particularly those of MPO and TGF-β1, were seen in the four drug groups. For example, the Nin group decreased by 58% (from 150.90 ± 9.62 to 63.37 ± 7.22 mg/g protein), while the carotene group only decreased by 8% (from 150.90 ± 9.62 to 138.80 ± 18.04 mg/g protein); and there was no significant difference in MPO between drug groups and the control group.

### 2.4. Effects on Antioxidant Enzymes of TRF

The antioxidant levels of GSH, NO, MDA, CAT and SOD in the lung tissue were also determined, as shown in Figure 4. As with the proinflammatory cytokine influx, the levels of antioxidant enzymes were significantly changed by the administration of BLM. Both the nutritional intervention groups (TRF, carotene and TRF + carotene groups) and drug groups were found to exert certain antioxidant effects and neither was necessarily better than the other. For instance, the CAT and GSH values of the drug groups were lower than those of the nutritional intervention groups, in which even the TRF-group’s values were increased to 5.75 ± 0.50 and 15.44 ± 1.86, respectively. That is to say, in these two indicators, the nutritional intervention groups even showed stronger antioxidation than the drug groups.

### 2.5. Effects on Histological Evaluation of TRF

HE staining was used to identify inflammation, while Masson’s trichrome staining was applied to examine the deposition of collagen in different sections of the tissue samples. A quantitative scoring system was used to evaluate the extent of PF and the severity of lung injury (as noted in Figure 5 and Figure 6). However, due to the large number of interventions, whether it is necessary to add them to the title remains to be discussed; the HE and Masson’s trichrome staining, lung specimens from the control group revealed ideal, intact pulmonary architecture with no evidence of tissue injury or fibrosis, nor any inflammation, hemorrhage, edema or emphysema. With the exception of the control group, lung architecture was distorted by BLM intervention to varying degrees in the other groups in which the alveoli had begun to obviously expand, the pulmonary interstitium had begun to bleed in varying degrees, and the alveolar septum was narrowed and breaking. The intervention groups all showed a certain therapeutic effect; however, according to the results, the effects on the drug groups were more distinct than those of the nutritional intervention groups. The results of the Masson’s staining were similar to those of the HE staining, indicating that the states of inflammation in the groups was consistent with the extents of their fibrosis.

### 2.6. IHC Determination on Collagen I and Collagen II of TRF

The excessive deposition of extracellular matrix (ECM) is a key feature of fibrosis, and the increased depositions of collagen I and II are one of the makers of PF [25,26]. In our study, the expression levels of collagen I and II were observed via IHC. Increased amounts of both type I collagen (yellow) (Figure 7) and type II collagen (yellow) (Figure 8) were detected, and were noted to be deposited in a disordered pattern in areas of fibrosis in model group. While lung tissues from the control group did not show depositions (Figure 7 and Figure 8, control group), TRF, carotene, Nin and Pir treatments all reduced the deposition of both types of collagens in the BLM groups. The reduction of collagen deposition in the nutritional intervention groups was found to be far less than in the drug groups, and the picture magnified 400 times was clearer and more visible. Our results thus suggest that TRF and carotene could treat BLM-induced PF in rats by decreasing the production of collagen I and II to a certain degree, and this finding concurs with the histological evaluation results.

### 2.7. Effects on TGF-β/Smad Signaling Pathway of TRF

Macrophage chemotaxis was reportedly rendered within a lesion via TGF-β signal activation, the activation and proliferation of fibroblasts were induced, the synthesis of collagen was increased, the expression of a great quantity of fibrotic cytokines and proinflammatory factors was stimulated, and the fibrotic response was further enhanced and sustained [27]. In this study, the expressions of the TGF-β1 protein, Smad2 protein, Smad3 protein and α-SMA protein were significantly (*p* < 0.05) downregulated compared to those in the control rats, while Smad7 protein was upregulated. In the WB group diagrams (Figure 9a–f), although there is no evidence of the statistical differences in the histograms (Figure 9c), and the values of the drug group were lower than those of the nutrition intervention groups overall. In the nutritional intervention groups, the treatment effect of the combined treatment group (TRF + carotene) was obviously improved, while the value in the model group was reduced from 1.53 ± 0.17 to 0.43 ± 0.07, and that of the control group was reduced from 0.24 ± 0.10. At the same time, however, gene expression (Figure 9g–j) was found to be more pronounced, which was not completely consistent with the WB results. It is worth mentioning that the TRF values were always statistically different from those of the control group, but similar to those of the model group. Overall, the above findings indicate that carotene and TRF exerted certain antifibrotic abilities, albeit far lower than the drug groups, and the remarkable anti-pulmonary fibrosis mechanisms may be related to the TGF-β/Smad signal pathway.

### 2.8. Effects on PI3K/Akt/mTOR Signaling Pathway of TRF

The PI3K/Akt/mTOR signaling pathway is an important intracellular signal transduction pathway, the inhibition of which can promote autophagy and PF [28,29,30]. Compared with the expressions of PI3K and mTOR proteins in the control group, those in the drug groups were significantly upregulated (*p* < 0.05), while their downregulation in the drug groups were more pronounced than those in the nutritional intervention groups. The RT-qPCR results (Figure 10e–g) showed that the expressions of PI3K mRNA, Akt mRNA and mTOR mRNA in the BLM-induced rats had increased significantly and were obviously downregulated in the drug groups (*p* < 0.05), while there were no statistical differences compared to the nutritional intervention groups in PI3K mRNA. Moreover, the protein results were confirmed at the genetic level. These findings suggest that, while TRF and carotene may alleviate PF through the PI3K/Akt/mTOR pathway to a certain extent, the treatment effect was not statistically significant (*p* > 0.05), and the expressions were more obvious at the gene level.

### 2.9. Effects on NF-κB Signaling Pathway of TRF

Nuclear factor kappa B (NF-κB) is widely present in inflammatory cells and, when stimulated to activation, takes part in a variety of pathological and physiological processes. The expression levels of inflammatory cytokines downstream of the NF-κB signaling pathway are especially relevant to the degree of inflammatory responses [31,32]. In this study, compared with the control group, the levels of p-p65 protein, p-IkBα protein, Ikkβ protein, TNF-α mRNA, IFN-γ mRNA, IL-13 mRNA, NF-κB mRNA, IkBα mRNA and Ikkβ mRNA were upregulated significantly (*p* < 0.05) in model group, especially in their gene expressions. Figure 11 shows that the phosphorylation of the NF-κB signaling pathway was decreased in all groups compared to the control and model groups, but with no significant differences (*p* > 0.05) between those affected groups. Furthermore, the results of RT-qPCR (also shown in Figure 11), were consistent with, but more obvious than, the WB results. It is worth noting that although the protein expressions of NF-κB in the lung tissue were downregulated in the nutritional intervention groups, and the activation of NF-κB inhibited, the level of downregulation was much lower than that of the drug groups.

### 2.10. Discussion

At present, few studies have reported the link between vitamin E and PF. Armutcu et al. illustrated that vitamin E plays a role in BLM-induced lung injury by reducing the inflammation caused by major inflammatory cytokines, such as IL-1β and IL-6, thereby providing a certain degree of protection and regulation [22]. Bese et al. reported that vitamin E supplementation immediately after irradiation could protect rats from radiation-induced PF [23], while Dede showed that vitamin E appeared to offset some of the severe lung damage in BLM-induced rats, as indicated by differences in serum concentrations [24]. In this work, the certain therapeutic effects of TRF and carotene on PF were revealed for the first time and compared with the effects of Nin and Pir. The main findings in our study are: (a) that TRF and carotene attenuate BLM-induced PF to a certain degree; (b) the mechanisms of TRF and carotene in the treatment of PF may be via the inhibition of TGF-β/Smad, PI3K/Akt/mTOR and NF-κB signaling pathways; (c) TRF and carotene have synergistic effects on some therapeutic indexes; and (d) the anti-inflammatory and antioxidant therapeutic effects of Nin and Pir in BLM-induced rats are similar.

The experiment described herein revealed a number of interesting results. In the previous studies on BLM model rats, the body weight generally decreased significantly, which lasted for a long time and recovered to varying degrees after the administration of different interventions [33,34]. However, in this paper, the body weight of the rats after modeling rose continuously after only a short weight loss period of 3–4 days. This weight change is similar to the experimental results of our prevention group [35]. The main difference between the two experimental processes was their modeling methods. The traditional modeling methods of PF are all surgical [36,37]. However, in this study, the tracheal instillation method was employed [38], as it not only reduces mortality but also reduces modeling time. The pathological results showed that our modeling was successful, as the values on the scoring criteria of HE and Masson’s staining rose from 0.25 ± 0.16 and 0.37 ± 0.18 to 7.00 ± 0.19 and 7.63 ± 0.26, respectively, through the administration of BLM. The therapeutic effects of TRF and carotene were the same, and the degree of pathological reduction of the two drugs was also similar, while not all therapeutic effects were strengthened by the combination with TRF. For example, the score of the Masson’s staining was not significantly reduced by the addition of TRF to Nin or Pir (Figure 6). The immunohistochemistry results were similar to those of pathology. The collagen deposition was reduced to a certain extent in the nutritional intervention groups, but the total amount of collagen was still much higher than that in the drug treatment groups. Furthermore, while TRF alone did reduce collagen deposition to some extent, its combination with Nin and Pir did not reduce collagen deposition significantly compared to the single drug groups (Figure 7 and Figure 8). In previous studies, the content of HYP in lung tissues has been noted as a quantitative index of fibrosis [39]. According to this indicator (Figure 2a), it was found that TRF, carotene, Nin and Pir alone could have a certain effect on the treatment of PF. However, when the carotene, Nin and PIR were combined with TRF, the values decreased and they played synergistic roles. These findings indicate that the addition of TRF to some drugs may reduce the symptoms of PF and strengthen treatment effects.

Recent studies show that PF can be prevented and treated by relevant natural products via the amelioration of oxidative stress, inhibition of inflammation and regulation of EMT, with mechanisms involving the p38 MAPK, TGF-β1/Smad, Nrf2-Nox4, NF-κB, PI3K/Akt and AMPK signaling pathways [40,41]. Our previously published experiments with a prevention group [35] showed that TRF and carotene can improve the prevention of PF through the TGF-β1/Smad, PI3K/Akt and NF-κB pathways. Nin is known to have the potential to inhibit the protein kinases involved in several molecular pathways and PI3K/AKT, JAK/STAT, TGF-β, WNT/β-catenin, VEGF and MARK are commonly reported in researches [42,43]. Moreover, numerous studies have reported the antifibrosis mechanisms of Pir, most of which involve TGF-β related genes and pathways [44,45,46]. Consequently, the three abovementioned pathways (TGF-β1/Smad, PI3K/Akt and NF-κB) were selected in this to ascertain whether TRF can enhance the therapeutic effect on PF, and if so, what is the specific mechanism involved therein. It was found that all proteins and genes were upregulated or downregulated to varying degrees following different interventions (Figure 9, Figure 10 and Figure 11). Furthermore, the possible mechanism involved in the inflammatory response was found by downregulating the expression of the NF-κB protein and inhibiting the release of inflammatory downstream cytokines IFN-γ, IL-13 and TNF-α of the NF-κB signal pathway, inhibiting the TGF-β/Smad signaling pathway by downregulating the protein expressions of Smad2/3 and TGF-β1, upregulating Smad7, and suppressing the PI3K/AKT/mTOR pathway by downregulating the levels of the p-AKT, PI3K and mTOR proteins. These results indicate that Nin, Pir, TRF and carotene truly can improve or treat PF via these three pathways, although the up- and downregulation of Nin and Pir were found to be significantly greater than those of TRF and carotene. Following the incorporation of TRF, the improvement of the pathways was enhanced (although there was no significant difference in some proteins), which indicated that the mechanisms by which TRF enhances the effect of drug treatment on PF include the TGF-β1/Smad, PI3K/Akt and NF-κB signaling pathways. These results provide reason to believe that the combination of TRF with appropriate drugs offers good potential in the development of improved effective treatments for PF. However, due to the complexity of TRF, it has not yet been established whether the most effective of its components is indeed γ-tocotrienol and, therefore, further cell experiments are required. Population experiments could be carried out to further confirm whether the addition of TRF in PF treatment can reduce symptoms and improve therapeutic effects, thereby providing more possibilities in the treatment of PF.

## 3. Materials and Methods

### 3.1. Animals, Interventions and Experimental Design

The protocol was approved by the Animal Research Ethics Committee of Southeast University (animal ethics approval No. 20210113003). Male Sprague-Dawley (SD) rats (200–220 g, specific-pathogen free (SPF) grade, certification No.110324201104564173) were purchased from SPF (Beijing, China) Biotechnology Co., Ltd. (Beijing, China). The rats were housed in cages at room temperature (23 ± 2 °C) with humidity 60 ± 10% and maintained in a 12:12 h light-dark cycle. Water and forage were freely available. TRF and carotene (natural mixed-carotene complex 20% oil concentrate) samples were acquired from the Malaysian Palm Oil Board (MPOB). The TRF and carotene were purified from palm oil through molecular distillation technology and their compositions are presented in Table 1 and Table 2 herein. Nin was purchased from Haiwo Technology Co., Ltd. (Jinan China); Pir was purchased from Bide Medicine Co., Ltd. (Shanghai, China); and bleomycin (BLM) was obtained from Thermo Fisher Scientific (Waltham, MA, USA). The 135 SD male rats were randomly divided into nine groups for the experiment, namely a control group, model group (BLM group), TRF group (200 mg/kg/D), Nin group (100 mg/kg/D), Pir group (100 mg/kg/D), carotene group (10 mg/kg/D), a TRF (200 mg/kg/D) and carotene (10 mg/kg/D) combined intervention group, a TRF (200 mg/kg/D) and Nin (100 mg/kg/D) combined intervention group and a TRF (200 mg/kg/D) and Pir (100 mg/kg/D) combined intervention group. Following one week of adaptive feeding, PF modeling via tracheal instillation of bleomycin was performed on all rats except for those in the control group, according to previously reported modeling methods [37]. After 28 days of modeling, the abovementioned interventions were dissolved in 0.5% sodium carboxymethyl cellulose (CMC-Na) and administered to the corresponding groups, while the rats in the control and model (BLM) groups were intragastrically administered with equal volumes of 0.5% CMC-Na for 28 days. Body weight and feed intake were measured daily to monitor changes in growth. All rats were euthanized and their corresponding organs were collected for analysis on the 56th day (28 days of modeling and 28 days of intervention).

### 3.2. Enzyme-Linked Immunosorbent Assay (ELISA)

Transforming growth factor beta 1 (TGF-β1), interleukin-6 (IL-6), myeloperoxidase (MPO), interleukin-1β (IL-1β) and tumor necrosis factor-alpha (TNFα) hydroxyproline (HYP) and MMP-7 ELISA kits were acquired from Nanjing Jin Yibai Biological Technology Co., Ltd. (Nanjing, China), while catalase (CAT), malondialdehyde (MDA), glutathione (GSH), nitric oxide (NO) and superoxide dismutase (SOD) were purchased from Nanjing Jiancheng Technology Co., Ltd. (Nanjing, China). On day 56, all of the abovementioned levels in the lung tissues were analyzed using the ELISA Kits. The standards supplied with the kits were used to generate the standard curve, according to the manufacturers’ protocols.

### 3.3. Histopathological Analysis

Hematoxylin-eosin (HE) and Masson’s trichrome staining were performed according to the manufacturer’s instructions. The lung tissues were dehydrated in graded alcohol, then fixed with paraffin blocks and cut into 4 μM flakes. The slices were dewaxed with xylene, rewatered with ethanol and washed with distilled water. Thereafter, they were stained with hematoxylin for 5 min, eosin for 2 min or the Masson compound dyeing solution for 5 min and, finally, with the bright green solution for 5 min. Routine dehydration, tissue removal and sealing were then carried out. Scoring criteria were based on those in a previously published report [47].

### 3.4. Immunohistochemical (IHC) Determination of Collagen I and Collagen II

Fresh lung tissue blocks, sized smaller than 0.5 cm × 0.5 cm × 0.1 cm, were washed with phosphate-buffered saline (PBS), fixed with 4% paraformaldehyde, soaked in ethanol, and then embedded in a stainless-steel mold. The tissue blocks were then cut to a thickness of 5 μm and immersed in xylene overnight. After dewaxing and the addition of water, they were washed with citric acid buffer for 20 min for antigen repair, then extracted with peroxide for 10 min and washed thrice with PBS solution for 5 min each time. Normal goat serum blocking solution was added dropwise on the slices for 20 min at room temperature, whereafter collagen I (AF7001, Affinity Biosciences, Beijing, China) and collagen II antibodies (AF0135, Affinity Biosciences, Beijing, China) were added dropwise, respectively; the slices were left at 4 °C overnight and then washed three times with PBS for 5 min each time. Next, biotinylated rabbit secondary antibody (SP-9001, ZSJQ-BIO, Guangzhou, China) was added to the slices and they were left for 20 min at 37 °C; streptomyces ovalbumin working solution (SA/HRP) labeled with horseradish enzyme was added to the slices, and they were left at 37 °C for 20 min. A diaminobenzidine (DAB) color kit was used for staining, after which the slices were, finally, counterstained and sealed. Image-Pro Plus 6.0 software (Rockville, MD, USA) was used to assess the results.

### 3.5. Western Blotting

Western blot (WB) analysis was performed according to the methods reported by Rong et al. [38]. Protein concentration, polyacrylamide gel electrophoresis, film rotation, closing, primary antibody incubation, film washing, secondary antibody incubation and enhanced chemiluminescence (ECL) development were all included in this experiment. The antibodies used were: anti-TGF-β1 (ab215715, Abcam (Cambridge, UK)), anti-Smad2 (ab40855, Abcam), anti-Smad3 (ab40854, Abcam), anti-Smad7 (25840-1-AP, Proteintech (Chicago, IL, USA)), anti-α-SMA (ab7817, Abcam), anti-PI3K (ab191606, Abcam), anti-p-Akt^ser473^ (66444-1-Ig, Proteintech), anti-mTOR (ab134903, Abcam), anti-p-IkBα (ab133462, Abcam), anti-p-P65 (ab76302, Abcam) and anti-p-IKKβ (AF3010, Affinity (Melbourne, Australia)).

### 3.6. Quantitative Reverse Transcription Polymerase Chain Reaction (qRT-PCR) Analysis

The methods of qRT-PCR analysis in this experiment were as described by Liu et al. [48], and included basic steps, such as the removal of residual genomic DNA, preparation of the reverse transcription reaction system, reverse transcription program settings, and fluorescence quantitative PCR. Primer information is detailed in Table 3.

### 3.7. Statistical Analysis

The data values were presented as the mean ± standard error of the mean (±SEM). One-way analysis of variance was applied for the significance and post-multiple comparison was assessed by Dunnett’s *t*-test (*p*-values of <0.05 were considered significant). GraphPad Prism 9.0 software (San Diego, CA, USA) was used to assess statistical significance.

## 4. Conclusions

In summary, our study demonstrated that TRF, carotene, Nin and Pir can all ameliorate PF by reducing inflammation and resisting oxidative stress via the TGF-β1/Smad, PI3K/Akt and NF-κB signaling pathways. Notably, when combined with TRF, the therapeutic effects of Nin and Pir on PF were enhanced. Thus, TRF may be a promising combination therapy for the treatment of PF in the future.

## Figures and Tables

**Figure 1 ijms-23-14331-f001:**
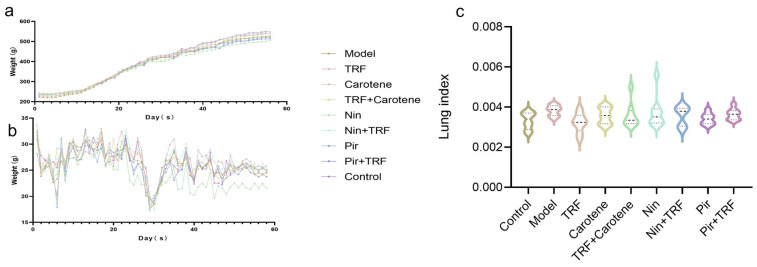
Effects on weight (g) and feed intake (g) of TRF: (**a**) Body weight changes in rat groups recorded every other day during the experiment (*n* = 15); (**b**) Feed intake changes in rat groups recorded every other day during the experiment (*n* = 15); (**c**) Lung/body weight ratios of rats on the last day of the experiment (*n* = 15). Data are expressed as the mean ± SD. There were no significant differences between the groups.

**Figure 2 ijms-23-14331-f002:**
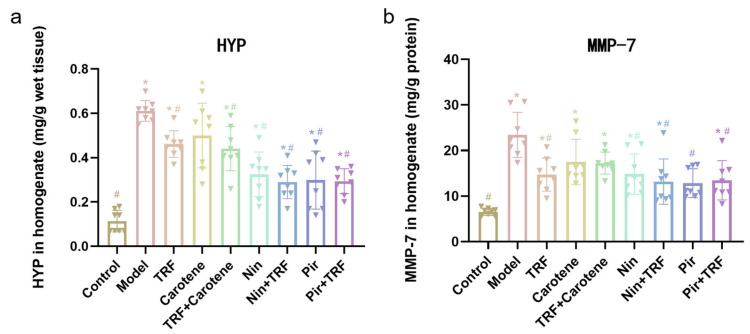
Effects of TRF and carotene on HYP (**a**) and MMP-7 (**b**) levels of TRF in lung tissue, *n* = 8. Note: * represents *p* < 0.05 in comparison to the control group; # represents *p* < 0.05 in comparison to the model group.

**Figure 3 ijms-23-14331-f003:**
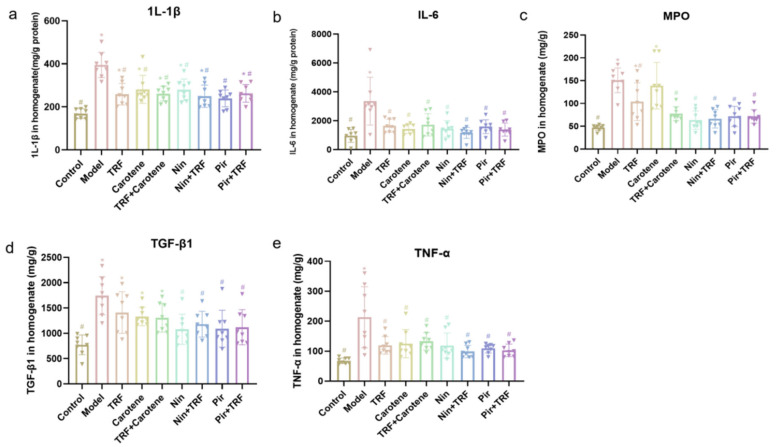
Effects on proinflammatory cytokine levels of TRF in the lung tissues of BLM-induced lung fibrosis (**a**–**e**); *n* = 8. Note: * represents *p* < 0.05 in comparison to the control group; # represents *p* < 0.05 in comparison to the model group.

**Figure 4 ijms-23-14331-f004:**
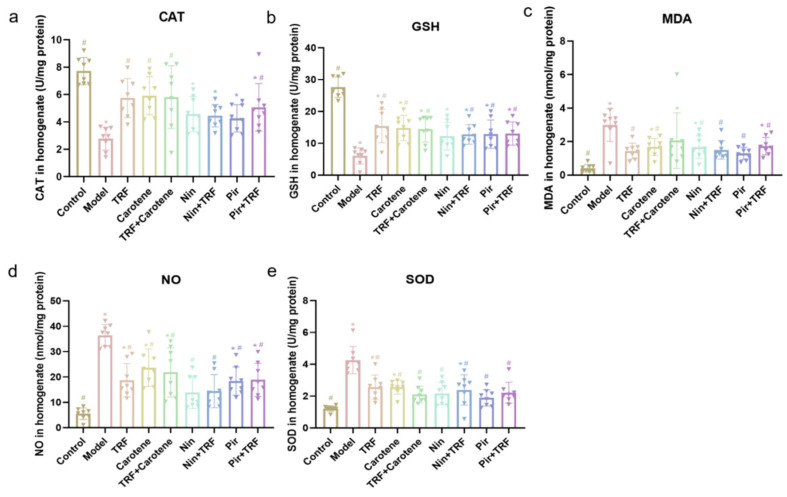
Effects on antioxidant enzymes and oxidative stress markers of TRF in the lung tissues of BLM-induced lung fibrosis (**a**–**e**); *n* = 8; Note: compared with CON group, * represents *p* < 0.05; Compared with MOD group, # represents *p* < 0.05.

**Figure 5 ijms-23-14331-f005:**
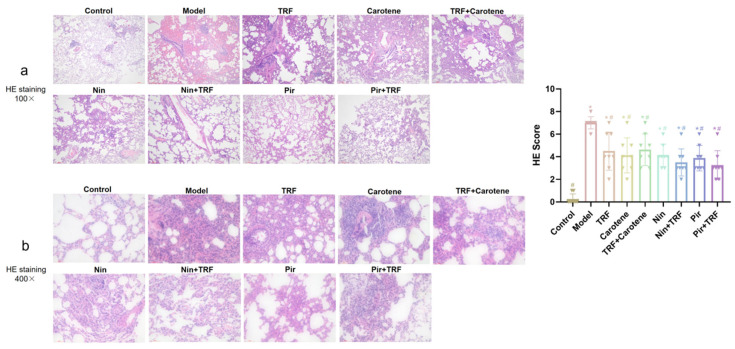
Effects on lung histopathological changes of TRF in H&E-stained specimens: Lungs from the different groups of rats were collected, and 4-μm thick sections were prepared and used for the assessment of tissue injury; (**a**) original magnification: 100×; (**b**) original magnification 400×; *n* = 8. Note: * represents *p* < 0.05 in comparison to the control group; # represents *p* < 0.05 in comparison to the model group.

**Figure 6 ijms-23-14331-f006:**
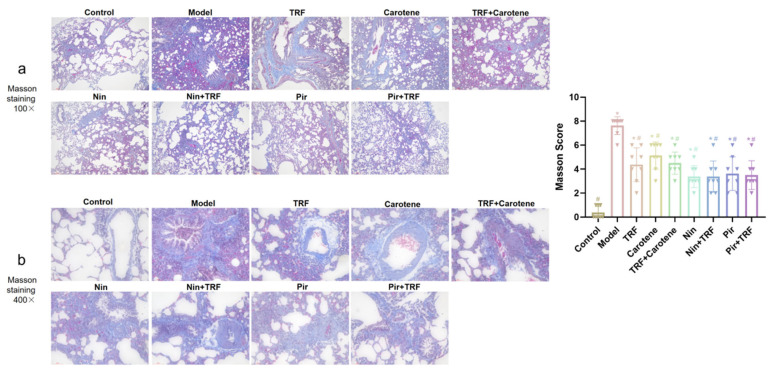
Effects on lung histopathological changes of TRF in Masson’s Trichrome stained specimens: Lungs from the different groups of rats were collected, and 4-μm thick sections were prepared and used for the assessment of tissue injury; (**a**) original magnification 100×; (**b**) original magnification 400×; *n* = 8. Note: * represents *p* < 0.05 in comparison to the control group; # represents *p* < 0.05 in comparison to the model group.

**Figure 7 ijms-23-14331-f007:**
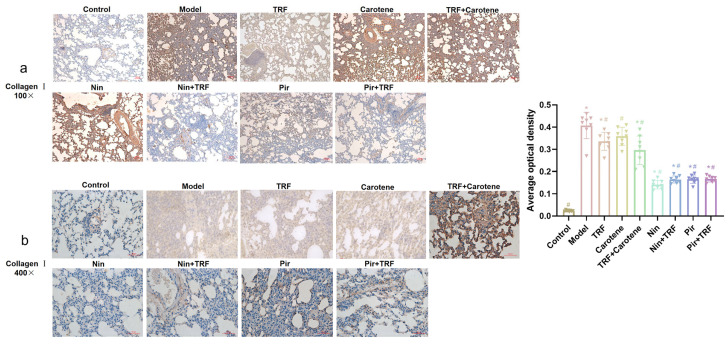
Amelioration of bleomycin-induced PF: Lungs from different rat groups were collected, and 5-μm thick sections were prepared and used for the assessment of tissue injury; representative images of IHC staining for collagen I in the lungs: (**a**) original magnification: 100×; (**b**) original magnification: 400×; *n* = 8. Note: * represents *p* < 0.05 in comparison to the control group; # represents *p* < 0.05 in comparison to the model group.

**Figure 8 ijms-23-14331-f008:**
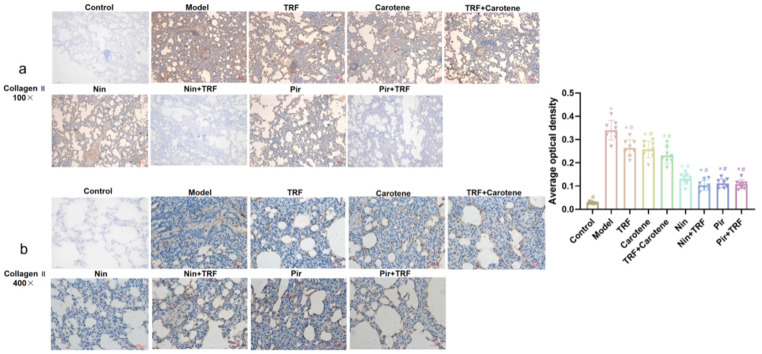
Amelioration of bleomycin-induced PF: Lungs from different groups were collected, and 5-μm thick sections were prepared and used for the assessment of tissue injury; representative images of IHC staining for collagen II in the lungs: (**a**) original magnification: 100×; (**b**) original magnification: 400×; *n* = 8. Note: * represents *p* < 0.05 in comparison to the control group; # represents *p* < 0.05 in comparison to the model group.

**Figure 9 ijms-23-14331-f009:**
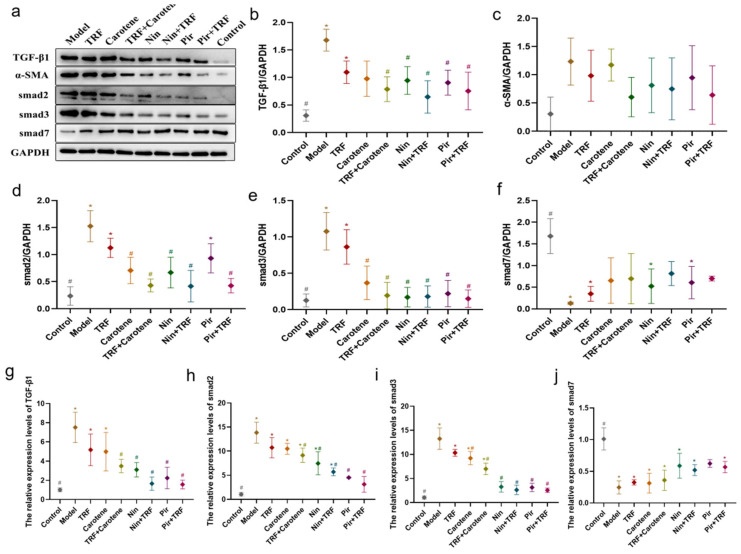
PF was protected by TRF via the inhibition of the TGF-β/Smad signaling pathway: WB analysis showed the expressions of fibrosis proteins induced by BLM in SD rats were decreased (**a**–**f**), as seen in TGF-β1 (**b**), Smad2 (**c**), Smad3 (**d**), Smad7 (**e**) and α-SMA (**f**) in the lung samples; *n* = 3; (**g**–**j**) Representative statistical analysis of TGF-β1 (**g**), Smad2 (**h**), Smad3 (**i**) and Smad7 (**j**) in mRNA by RT-qPCR; *n* = 3. Note: * represents *p* < 0.05 in comparison to the control group; # represents *p* < 0.05 in comparison to the model group.

**Figure 10 ijms-23-14331-f010:**
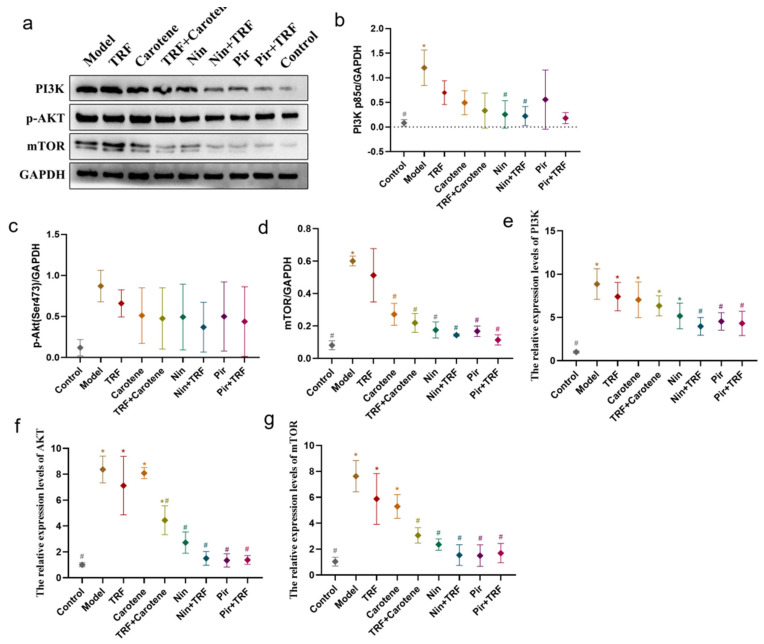
Inhibition of PI3K/AKT/mTOR signaling pathway provides protection against PF: (**a**–**d**) The expression of fibrosis proteins induced by BLM in SD rats was decreased; the protein expressions of PI3K (**b**), p-AKT (**c**) and mTOR (**d**) in lung samples were examined by WB analysis; *n* = 3. (**e**–**g**) Representative statistical analysis of PI3K (**e**), AKT (**f**) and mTOR (**g**) mRNA by RT-qPCR; *n* = 3. Note: * represents *p* < 0.05 in comparison to the control group; # represents *p* < 0.05 in comparison to the model group.

**Figure 11 ijms-23-14331-f011:**
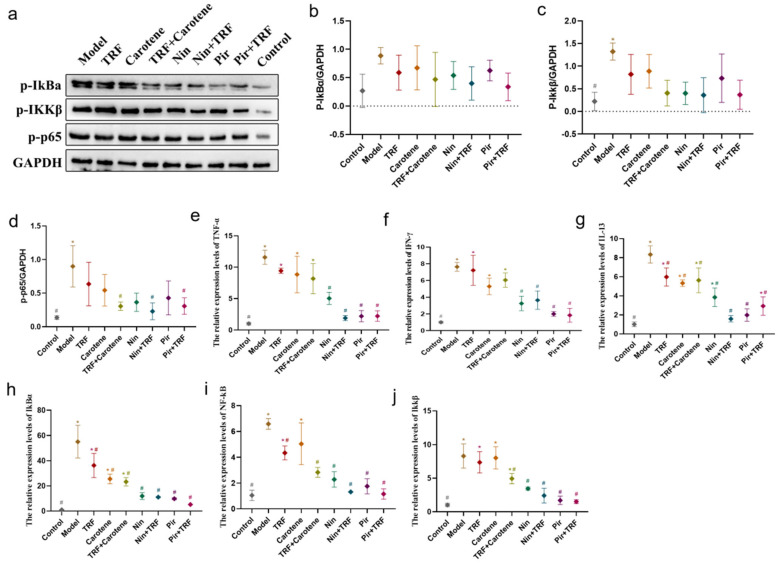
PF was protected by TRF via the inhibition of the NF-κB signaling pathway: (**a**–**d**) The expression of fibrosis proteins induced by BLM in SD rats was decreased; protein expressions of p-IkBα (**b**), Ikkβ (**c**) and p-p65 (**d**) in lung samples were examined by WB analysis; *n* = 3. (**e**–**j**) Representative statistical analyses of TNF-α (**e**), IFN-γ (**f**), IL-13 (**g**), IkBα (**h**), NF-κB (**i**) and Ikkβ (**j**) in mRNA by RT-qPCR; *n* = 3. Note: * represents *p* < 0.05 in comparison to the control group; # represents *p* < 0.05 in comparison to the model group.

**Table 1 ijms-23-14331-t001:** Composition of total mixed-carotene complex 20% oil concentrate (carotene).

Components	Values (mg/g)
α-Carotene	65
β-Carotene	135
γ-Carotene	0.5
Lycopene	0.1
Total mixed-carotene complex	200.6

**Table 2 ijms-23-14331-t002:** Composition of tocotrienol-rich fractions (TRF).

Components	Values (wt/wt)
α-Tocopherol	12.5
α-Tocotrienol	12.8
β-Tocotrienol	2.0
γ-Tocotrienol	19.5
δ-Tocotrienol	5.5
Total mixed tocotrienols	39.8
Tocotrienol/Tocopherol complex	52.3

**Table 3 ijms-23-14331-t003:** Primer information for qRT-PCR analysis.

mRNA	Forward Primer	Reverse Primer
TGF-β1	TCGCCCTTTCATTTCAGAT	TTTGCCGATGCTTTCTTG
Smad2	AGGTGTCTCATCGGAAAG	CTCTGGTAGTGGTAAGGGT
Smad3	AGCTTACAAGGCGGCACA	TGGGAGACTGGACGAAAA
Smad7	CTTCCTCCGATGAAACCG	TCGAGTCTTCTCCTCCCAGTA
PI3K	GAAACCCAGTCACCTAGGGC	GGTGGGCAGTACGAACTCAA
AKT	GAGGAGCGGGAAGAGTG	GTGCCCTTGCCCAGTAG
mTOR	GGTGGACGAGCTCTTTGTC	AGGAGCCCTAACACTCGGAT
TNF-α	TGAGCACAGAAAGCATGATC	CATCTGCTGGTACCACCAGTT
IFN-γ	TTGCAGCTCTGCCTCAT	TTCGTGTTACCGTCCTT
IL-13	CTCGCTTGCCTTGGTGG	TGATGTTGCTCAGCTCCTC
NF-κB	CTGTTTCCCCTCATCTTTCC	GTGCGTCTTAGTGGTATCTGTG
IkBα	CCAACTACAACGGCCACA	CAACAGGAGCGAGACCAG
Ikkβ	CATTGTTGTTAGCGAGGAC	CCCTTTGCCGAGGTTGC
GAPDH	AAGAAGG TGGTGAAGCAGGC	TCCACCACCCT GTTGCTGTA

Note: TGF-β1, transforming growth factor beta 1; Smad2, Smad2; Smad3, Smad3; Smad7, Smad7; PI3K, phosphatidylinositol 3-kinase; AKT, (protein kinase B, PKB); mTOR, mammalian target of rapamycin; TNF-α, tumor necrosis factor-alpha; IFN-γ, interferonγ; IL-13, interleukin-13; NF-κB, nuclear factor kappa-B; IkBα, NF-kappa-B inhibitor alpha; Ikkβ, inhibitor of nuclear factor kappa-B kinase.

## Data Availability

Not applicable.

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
