# Peer review of "Tocotrienol-Rich Fractions Offer Potential to Suppress Pulmonary Fibrosis Progression"

_ijms, 2022, doi:10.3390/ijms232214331_

Round 1
Reviewer 1 Report
The study demonstrated that TRF, carotene, Nin and Pir ameliorated PF by reducing inflammation and resisting oxidative stress. And it is pointed out that TRF may be a promising combination therapy for the treatment of PF in the future. But there are major problems need to be fixed in the manuscript.
1. The lung/body weight ratio is not enough to confirm pulmonary functions, forced vital capacity (FVC) and dynamic compliance (Cdyn) should be monitored.
2. NF-κB detected to verify the change of cytokines, and oxidative stress regulated by PI3K/AKT/mTOR. However, none of the regulatory mechanisms were cleared and completed, just presented on the superficial. Signaling pathway inhibitors and ROS regulators (NAC/H2O2) should be performed in this work.
3. The therapeutic effect of Nin alone seemed to be better than that of TRF, and even the TRF+Nin group was not more significant than Nin group in this study. Thus, the research of TRF seemed to be valueless compared with Nin, which has been used as a drug of PF treatment. So why you chosen and concluded TRF as a potential drug for PF?
4. Grammatical errors and contextual cohesion problems should be greatly proofread and modified in the article.
5. Minor problems should be corrected in the manuscript:
a. Fig. 10 was uploaded twice, total-AKT and p-mTOR should be detected.
b. The quality and resolution of IHC pictures need to be improved.
c. Scale bars should be added to all microscopic pictures.
d. In tissue protein detection, author chosen only one sample in each group, it will be better to detect at least three samples.
e. The panel size of Figure 3b should normalized to other panels.
Reviewer 2 Report
The paper is very interesting in the field. Authors presented a study describing the potential effect of TRF and carotene when they are administered with Pir and Nin. The paper is well written and the report is sound, but I do think that minor issues need to be dealt with prior to publication:
Concerns:
1. Please change, in the abstract, intratracheal injection for “intratracheal instillation”
2. Line 125, change mode for “model”
3. Please in the subheading results “3.2 ffects on HYP and MMP-7 levels”, indicate the “effects” on what? Apply for the all subheadings and legend of figures.
4. Please describe the score criteria of the histological analysis.
5. I am not pretty sure if the term “nutritional intervention” being the adequate
6. Figure 10 is duplicated
